# Two New Components from an Association of Marine Sponges *Poecillastra* sp. and *Jaspis* sp. and Their Inhibitory Effects on Biomarkers for Benign Prostatic Hyperplasia

**DOI:** 10.3390/md21090491

**Published:** 2023-09-14

**Authors:** Buyng Su Hwang, Sangbum Lee, Eun Ju Jeong, Jung-Rae Rho

**Affiliations:** 1Bio-Resource Industrialization Center, Nakdonggang National Institute of Biological Resources, Sangju 37242, Republic of Korea; hwang1531@nnibr.re.kr; 2Department of Oceanography, Kunsan National University, Kunsan 54150, Republic of Korea; sblee08@kunsan.ac.kr; 3Department of GreenBio Science, Gyeongsang National University, Jinju 52725, Republic of Korea

**Keywords:** marine sponge, (2*S*, 3*S*)-capreomycidine, tramiprosate, *J*BCA, ECD comparison, 5-alpha reductase type 2, benign prostatic hyperplasia (BPH)

## Abstract

Benign prostatic hyperplasia (BPH), characterized by the enlargement of the prostate gland and subsequent lower urinary tract symptoms, poses a significant health concern for aging men with increasing prevalence. Extensive efforts encompassing in vitro and in vivo models are underway to identify novel and effective agents for the management and treatment of BPH. Research endeavors are primarily channeled toward assessing the potential of compounds to inhibit cell proliferation, curb inflammation, and display anti-androgenic activity. Notably, through screening aimed at inhibiting 5-alpha reductase type 2 (5αR2) in human prostatic cells, two acyl compounds (**1** and **2**) were isolated from a bioactive fraction sourced from an association of marine sponges *Poecillastra* sp. and *Jaspis* sp. The complete structure of **1** was determined as (*Z*)-dec-3-enony (2*S*, 3*S*)-capreomycidine, ascertained by JBCA and ECD comparison. While the absolute configurations of **2** remained unassigned, it was identified as a linkage of a 2, 7*S**-dihydoxy-9*R**-methyloctadecanoyl group with the 2-amino position of a tramiprosate moiety referred to as homotaurine. Evaluation of both compounds encompassed the assessment of their inhibitory effects on key biomarkers (5αR2, AR, PSA, and PCNA) associated with BPH in testosterone propionate (TP)-activated LNCap and RWPE-1 cells.

## 1. Introduction

Prostatic ailments, including conditions such as prostate cancer and benign prostatic hyperplasia, arise due to the influence of multifarious determinants encompassing age, hormonal profiles, and genetic predisposition. Benign prostatic hyperplasia (BPH), a prevalent medical condition among older males, affects around 60% of individuals in their sixth decade of life. BPH involves the non-malignant expansion of the prostate gland, leading to its enlargement and causing lower urinary tract symptoms (LUTS), as well as obstructive manifestations at the bladder outlet (BOO). While BPH is relatively uncommon prior to the fourth decade of life, approximately half of men experience symptoms associated with BPH by the age of 50. The incidence of BPH increased by 10% per decade, reaching an estimated prevalence of 80% in the octogenarian phase [1,2]. Despite exhaustive research efforts, certain etiological facets of BPH remain concealed. Among the predominant hypotheses, the interplay of age-related changes and disruptions in hormonal balance stands as the principal foundation in the pathogenesis [3].

Diverse botanical treatments are utilized in both traditional and alternative medical practices to alleviate symptoms related to the lower urinary tract. Among these botanical interventions, *Serenoa repens* (SR)*, Pygeum africanum*, and *Urtica dioica* stand out prominently [4,5,6]. Preclinical investigations highlight the multifaceted mechanisms through which the SR extract exerts its effects. These mechanisms include inhibiting the binding of dihydrotestosterone (DHT) receptors within the cytosol of prostatic cells, and reducing the expression and/or activity of 5-alpha reductase, cyclooxygenase, and 5-lipoxygenase. Furthermore, SR induces apoptotic responses in prostatic epithelial cells, demonstrates antiestrogenic attributes, acts as a spasmolytic agent by impeding calcium channel activity, and counteracts beta-adrenergic actions [4]. These combined actions have spurred clinical and experimental research into the potential of SR as a therapeutic approach to alleviate symptoms associated with benign prostatic hyperplasia.

Various in vitro and in vivo models have been developed to find promising new substances for managing and treating BPH. Research primarily focuses on evaluating properties like cell proliferation inhibition, anti-inflammatory, anti-androgenic activity, and antioxidant properties. In our ongoing search for new sources of natural products that can effectively manage BPH, we found that an extract prepared from an association of marine sponges *Poecillatra* sp. and *Jaspis* sp. exhibited potent activity on 5-alpha reductase in human prostatic cells, surpassing the efficacy of the most commonly used 5α-reductase inhibitor, finasteride. Marine sponges of the *Poecillatra* and *Jaspis* genera are widely recognized as rich sources of bioactive compounds of diverse skeletal classes [7,8]. The association of *Poecillatra* sp. and *Jaspis* sp. has also yielded novel metabolites, including anti-angiogenic bis(dihydroxystyrenyl) imidazole alkaloids [9], dihydroxystyrenes [10] and cytotoxic bromotyrosine derivatives [11]. By activity-guided fractionation, we successfully isolated two new compounds, designated as **1** and **2**, from the association of *Poecillatra* sp. and *Jaspis* sp. In this report, we provide detailed information about the isolation and structural determination of compounds **1** and **2**, along with their therapeutic effects on the expressions of 5-alpha reductase type 2 (5αR2), androgen receptor (AR), prostate-specific antigen (PSA), and proliferating cell nuclear antigen (PCNA) in testosterone propionate (TP)-activated LNCap and RWPE-1 cells.

## 2. Results and Discussion

### 2.1. Structural Determination of Compounds 1 and 2

Compounds **1** and **2** were detected through MS scanning in the negative mode for the bioactive fraction (30% H_2_O and 70% MeOH) of the preserved extract from the association of *Poecillastra* sp. and *Jaspis* sp. They were subsequently isolated by using reversed silica HPLC. Both compounds possessed a linear carbon chain. The structures of these isolated compounds were elucidated through a combination of 1D and 2D NMR experiments, along with MS data (Figure 1). 

Compound **1** was identified to have the molecular formula C_16_H_28_N_4_O_3_ based on a peak ([M − H]^−^ = 323.2087, ∆ = 0.5 ppm) in the negative HR-ESIMS and the ^13^C NMR spectra. The broad UV absorption band appeared at 215 nm, and the IR spectrum displayed absorption peaks at 1626 and 3315 cm^−1^, indicating carbonyl and hydroxy groups. In the ^1^H NMR spectrum of **1**, two olefinic protons were identified at δ_H_ 5.55 and 5.61, along with five protons within the mid-chemical shift range (δ_H_ 3.08, 3.23, 3.40, 3.75, and 4.39), three shielded protons (δ_H_ 1.75, 1.90, and 2.10), several aliphatic protons in the range of δ_H_ 1.26 to 1.36, and one methyl at δ_H_ 0.89. The ^13^C NMR spectrum revealed sixteen resonances, which included two carbonyl carbons (δ_C_ 174.5 and 174.7), two olefinic carbons (δ_C_ 122.8 and 135.1), one non-protonated carbon (δ_C_ 155.5), two methines (δ_C_ 57.6 and 53.6), eight methylenes (δ_C_ 23.2, 23.7, 28.4, 30.1, 30.5, 32.9, 35.6, and 38.4), and one methyl (δ_C_ 14.4), as confirmed by the HSQC spectrum.

The analysis of COSY and TOCSY spectra led to the identification of two fragments, as shown in Figure 2A. One of these fragments was deduced to be a linear hydrocarbon chain. The second fragment contained four protons with chemical shift values within the mid-range, which were indicative of nitrogen-bearing carbons. Among these protons, H-3 and H-5 exhibited common correlations with a non-protonated carbon at δ_C_ 155.5 (C-6) in the HMBC spectrum, suggesting the presence of a pyrimidine moiety. Furthermore, the proton at δ_H_ 4.39 (H-2) showed a broad cross-peak in the HMBC spectrum, which implies correlations with two nearby carbonyl carbons (δ_C_ 174.5 and 174.7). Based on the HMBC correlation between δ_H_ 3.75 and δ_C_ 174.5, it was deduced that the proton at δ_H_ 4.39 (H-2) was connected to the carboxylic carbon at δ_C_ 174.5 (C-1) through a two-bond linkage. The existence of a carboxylic group was supported by a characteristic band (1626 and 3315 cm^−1^) observed in the IR spectrum. Furthermore, H-2 revealed a connection to the carbonyl carbon in the amide group. The established partial structure was identified as a cyclic arginine-derived non-proteinogenic amino acid. The proton at δ_H_ 3.08 (H-2′) was determined to be connected to the carbonyl carbon at δ_C_ 174.7, confirmed by the HMBC correlation of H-2′/C-1′. The position of the double bond was supported by the HMBC correlations of H-3′/C-2′, H-4′/C-2′, H-4′/C-5′ and H-4′/C-6′. The geometry of the double bond in the chain was confirmed as the *Z*-form by the observed NOE correlation between H-2′ and H-5′. Lastly, considering the molecular formula, a hexane chain was identified as linked to the olefinic group. Consequently, the structure of **1** was elucidated as featuring a dec-3-enoyl group attached to the 2-amino position of the cyclic arginine-derived non-proteinogenic amino acid. 

The configuration of the two chiral centers within **1** was achieved through *J*BCA and ECD comparisons. Initially, the relative configurations of C-2 and C-3 were established using the homo-/heteronuclear coupling constants measured by the ^1^H and HETLOC NMR spectra presented in Figure 3. The moderate homonuclear coupling constant value (^3^*J*_HH_ = 6.4 Hz) suggested the presence of exchangeable rotation around the C-2 and C-3 bond, while the large absolute heteronuclear coupling value (^2^*J*_CH_ = −5.3 Hz) between H-3 and C-2 indicated a *gauche* relationship between H-3 and the amine group on C-2. The small absolute value between H-2 and C-3 indicated a portion of the *anti*-relationship between H-2 and the amine group on C-3. Consequently, the configurations of C-2 and C-3 were identified as 2*S*^*^ and 3*S*^*^, respectively. The absolute configuration of **1** was established by comparing the measured and calculated ECD spectra. To save time, the ECD spectrum was calculated for a partial structure truncated at C-5′. As depicted in Figure 4, the calculated ECD spectrum for the 2*S*3*S* isomer closely matched the experimental one. The cyclic arginine-derived non-proteinogenic amino acid in **1** was recognized as (2*S*, 3*S*)-capreomycidine (epicapreomycidine) [12]. 

Compound **2** was determined to have the molecular formula C_22_H_45_NO_6_S based on the negative HR ESIMS and the ^13^C NMR spectrum, indicating one unsaturation degree. The IR spectrum showed characteristic peaks at 1188 and 1061 cm^−1^, corresponding to a sulfate or sulfonic acid group. The MS/MS data for **2** revealed three major fragments *m*/*z* 80, 138, and 166. The IR peaks and molecular fragments indicated the presence of a sulfonic acid. The ^1^H NMR spectrum displayed shielded signals for a linear carbon chain and four signals (δ_H_ 2.82, 3.33, 3.60, and 3.98) in the middle spectral region. The ^13^C NMR spectrum contained two methyl groups, sixteen methylenes, two oxymethines, one methine carbon, and one carbonyl carbon. Based on the provided information, **2** was deduced to possess a linear carbon chain structure with a sulfonic acid moiety. 

The COSY and TOCSY spectra provided insight into the structural components of two carbon moieties, as shown in Figure 2B: a propane unit and a linear carbon chain from well-defined proton connections. The chemical shifts of the terminal protons and carbons in the propane segment suggested the presence of amine and sulfonic acid groups. This deduction was corroborated by the mass fragment at *m*/*z* 138, confirming the identity as 3-aminopropane-1-sulfonic acid (3-APS) (Appendix A). Notably, this moiety was also recognized as tramiprosate (homotaurine), a compound under development as a potential treatment for Alzheimer’s disease [13]. The linkage of this moiety to carbonyl carbon at δ_C_ 177.6 (C-1′) was established through an HMBC correlation, and the oxymethine proton at δ_H_ 3.98 (C-2′) showed connectivity to the same carbonyl carbon. As shown in Figure 2B, the presence of the 2-hydroxy-4-methylpentane unit was revealed by the COSY and HMBC correlations. The position of the unit in the linear carbon chain was determined by the HMBC correlations of H-6′/C-4′ and H-6′/C-5′ and supported by the mutual correlations of both H-2′ and H-7′ with C-4′, C-5′, and C-6′ in the HSQC-TOCSY spectrum. Based on the molecular formula, the COSY and HMBC correlations, the remaining portion, excluding 3-APS, was identified as a 2, 7-dihydroxy-9-methyloctadecanoyl group. Unfortunately, the Mosher reaction for determining the configuration of C-2′ and C-7′ was unsuccessful. However, the relative configurations of C-7′ and C-9′ were deduced through *J* configuration analysis (*J*BCA), as depicted in Figure 5. The homo-/heteronuclear coupling constants were measured by DQFCOSY and HECADE spectra. Utilizing the non-equivalent proton positions on C-8′, the relative configurations of C-7′ and C-9′ were determined as 7*S** and 9*R**, respectively, based on heteronuclear coupling constants. Further support for this assignment arose from observing NOE cross-peaks of H-7′/H-10′, H-7′/H-19′, and H-8′a/H-19′ in the NOESY spectrum. Thus, compound **2** was elucidated to be 2′,7′(*S**)-dihydroxy-9′(*R**)-methyloctadecanoyl tramiprosate. 

### 2.2. Biological Activity of Compounds 1 and 2 

In order to access the therapeutic effects of **1** and **2** on BPH regulation, we measured the expressions of 5αR2, AR, PSA, and PCNA in LNCap and RWPE-1 cells. Before evaluating the activities of **1** and **2** in these prostate cell lines, we determine the cytotoxicity of the compounds using a CCK-8 assay. LNCap and RWPE-1 cells were exposed to **1** or **2** for 24 h, and cell viability was measured. As demonstrated in Figure 6, compounds **1** and **2** displayed negligible cytotoxicity against LNCap cells in a concentration range of up to 10 µM (cell viability > 98% of control). Compound **2** exhibited slight cytotoxicity against RWPE-1 cells at concentrations of 1 and 10 µM. Based on the MTT assay results, we expect that compound **2** may exhibit anti-proliferative activity. However, in this study, we did not perform additional experiments to precisely quantify this anti-proliferative effect. Although a mild decrease in cell viability was observed at a concentration of 10 μM for **2** in RWPE-1 cells, the reduction was less than 10%. Importantly, no toxicity was observed in LNCap cells. Consequently, concentrations of 1 and 10 μM were chosen for evaluating the effects of these compounds in both cell types. 

To evaluate the potential impact of **1** and **2** on BPH, we measured their inhibitory effects on the expression of 5α-reductase type 2 (5αR2). As shown in Figure 7, the expression of 5αR2 was induced by the treatment of TP in LNCap and RWPE-1 cells, and the pretreatment of cells with **1** or **2** (1 and 10 µM) significantly suppressed the 5αR2 expression. Particularly, compound **2** exhibited a more pronounced decrease in 5αR2 levels at the same concentration (Figure 7A,B). At the concentration of 10 µM, compound **2** decreased the expression level of 5αR2 protein induced by TP to 42% and 48% of TP-only treated cells in LNCap and RWPE-1 cells, respectively.

The transformation of testosterone into DHT, facilitated by 5αR2 in androgen-responsive target cells, is a fundamental process implicated in the development of BPH [14]. It is widely recognized that inhibitors of 5αR2, such as finasteride (Fina) or dutasteride, can impede BPH progress by suppressing DHT synthesis [14,15]. Additionally, 5αR2 is recognized as the primary target protein through which SR ameliorates symptoms related to BPH. Recent meta-analyses have demonstrated that the efficacy of SR is comparable to that of finasteride and tamsulosin, and notably superior to that of a placebo in treating mild to moderate lower urinary tract symptoms (LUTS) [16]. 

To assess the potential therapeutic effects of **1** and **2** on AR in prostate cells, we employed Western blot analysis to evaluate the levels of AR expression. As illustrated in Figure 8, the treatment with TP resulted in an increase in AR expression in both LNCap and RWPE-1 cells. In cells treated with **2**, a slight decrease in AR expression was observable in both cell lines, whereas the reduction induced by **1** did not reach statistical significance. Specifically, the expression of AR was reduced by **2** (10 µM) to 72% and 76% of the levels in TP-treated cells in LNCap and RWPE-1 cells, respectively.

Androgens and androgen receptors (AR), a member of the steroid receptor superfamily, are widely recognized as playing a significant role in the development of BPH [17]. In epithelial cells, AR is thought to contribute to BPH progression by orchestrating interactions between epithelial and stromal cells. These interactions involve changes in epithelial–mesenchymal transition, leading to subsequent stromal cell proliferation [18,19]. While blocking the androgen/AR signaling pathway has been shown to decrease BPH volume and alleviate lower urinary tract symptoms, the precise mechanisms through which androgen/AR signaling influences BPH development remain unclear [20,21]. 

Upon androgen activation, the androgen-activated AR dissociates from chaperones and translocates into the nucleus. The interaction of AR with androgen response elements (AREs) is recognized to trigger the expression of genes such as PSA and PCNA [22,23]. PSA is recognized as a critical androgen-regulated gene and acts as a specific marker for detecting prostate cancer, rendering it valuable for prostate cancer screening and evaluation. In LNCap cells stimulated by TP (Figure 9A), the application of **1** and **2** significantly reduced the expression of PSA at a concentration of 10 µM. The expression level of the PSA protein diminished to 78% and 47% of TP-only treated cells by **1** and **2**, respectively. In TP-induced RWPE-1 cells (Figure 9B), the expression of PCNA was attenuated by **2** at a concentration of 10 µM, while the induction of PCNA expression was observed in cells treated with **1**.

## 3. Materials and Methods 

### 3.1. General Experiment Procedures 

Optical rotations were measured on a JASCO P-1010 polarimeter with a 1 cm cell and circular dichroism spectrum was recorded on a JASCO J-1500 CD spectrometer (Jasco Corporation, Tokyo, Japan). The UV and IR spectra were acquired on a Varian Cary 50 and JASCO FT/IR 4100 spectrometers, respectively. The NMR spectra were measured on a Varian VNMRS 500 MHz spectrometer (Varian, Palo Alto, CA, USA) with a 3 mm ID probe in MeOH-*d*_4_ solvent, which was referenced by residual solvent peaks at δ_H_ 3.30 and δ_C_ 49.0). The high-resolution ESI mass spectrum was acquired by using a SCIEX X500R (Sciex Co., Framingham, MA, USA). The HPLC was performed using an Agilent 1200 system (Santa Clara, CA, USA) using Phenomenex polar C18 and YMC ODS-A columns and an ELSD detector. Quantum calculations were conducted by Dell PowerEdge R740 server (Dell, Round Rocks, TX, USA) installed Gaussian 16 (Gaussian Inc., Wallingford, CT, USA) and Spartan 20 (Wavefunction Inc., Irvine, CA, USA) software. 

### 3.2. Animal Material

A specimen of a mutualistic association of Poecillastra sp. and Jaspis sp. (No. 08K-3) was obtained through scuba diving at a depth of 20 m off the coast of Keomun Island, Korea in July 2008. The identification of the sample was confirmed by Professor Chung Ja Sim from Hannam University, Korea. Following collection, a methanolic extract of the specimen was stored in a refrigerator at −28 °C for research purposes. 

### 3.3. Isolation of Compounds 1 and 2 

The stored extract dissolved in MeOH was dried in vacuo and was partitioned between dichloromethane and distilled water. The organic layer was repartitioned with a mixture of n-hexane and 15% aqueous MeOH. Then, the aqueous MeOH layer was subjected to reversed-phase silica gel flash column chromatography eluting with solvents of decreasing polarity (MeOH/H_2_O = 5/5; 6/4; 7/3; 8/2; 9/1; 100% MeOH; 100% acetone) to give seven fractions (MR1~MR7). Guided by DHT inhibition assay, fraction MR3 (30 mg) was selected for further research. Compounds **1** and **2** were isolated by reversed-phase silica HPLC eluting with a gradient solvent system (A: H_2_O with 0.1% formic acid, B: 100% acetonitrile, from 20% to 100% B for 30 min) at retention times of 14.6 and 24 min, respectively. HPLC conditions consisted of a Phenomenex polar-C18 column 150 x 4.6 mm, 4 us, 1 mL/min, ELSD detector. Compound **1** was purified by using YMC ODS-A column under isocratic solvents of 45% ACN and 55% H_2_O to yield 2.5 mg. Compound **2** was purified by a solvent system of 60% MeOH and 40% H_2_O to yield 3.0 mg. 

Compound (**1**). Amorphous oil. ${\left[\alpha \right]}_{D}^{25}$ + 18.4 (*c* 0.2, MeOH). UV (MeOH) λ_max_ (log ε): 215 (3.1) nm. IR (film) ν_max_: 3315, 2924, 1626, 1464, 1378 cm^−1^. ^1^H (500 MHz) and ^13^C (125 MHz) NMR data, see Table 1. HR-ESI MS (negative-ion mode) *m*/*z*: 323.2087 [M−H]^−^ (calcd for C_16_H_28_N_4_O_3_, 323.2089).

Compound (**2**). Amorphous oil. ${\left[\alpha \right]}_{D}^{25}$ − 5.5 (*c* 0.1, MeOH). IR (film) ν_max_: 3348, 2924, 1630, 1463, 1188, 1061 cm^−1^. ^1^H (500 MHz) and ^13^C (125 MHz) NMR data, see Table 2. HR-ESI MS (negative-ion mode) *m*/*z*: 450.2887 [M−H]^−^ (calcd for C_22_H_45_NO_6_S, 450.2895).

### 3.4. ECD Calculation of Compound 1

For the purpose of reducing the calculation time, the structure of compound **1** truncated at C-5′ was used. The conformational searches of the two isomers (2*S*3*S* and 2*R*3*R*) for the truncated **1** were performed by Spartan 20 software (Wavefunction Inc., Irvine, CA, USA) following a protocol reported in the literature [24]. The selected conformers were reoptimized at the m062X/6-31G(d,p) level with CH_3_OH solvent by using Gaussian 16 software (Gaussian inc., Wallingford, CT, USA). The ECD spectra of the conformers corresponding to the two isomers were calculated to the B3LYP//6-31G(d,p) level. A Boltzmann distribution for the optimized conformers was used for calculated ECD spectra. 

### 3.5. Cell Cultures

The RWPE-1 cell line (normal human prostatic epithelial cells) and LNCap cell line (human prostatic adenocarcinoma cells) were obtained from the American Type Culture Collection. RWPE-1 cells were cultured in keratinocyte serum-free medium supplemented with 0.05 mg mL^−1^ bovine pituitary extract, 5 ng mL^−1^ epidermal growth factor, and 1% (*v*/*v*) antibiotics (100 U mL^−1^ penicillin and 100 μg mL^−1^ streptomycin). LNCap cells were cultured in RPMI1640 containing 10% (*v*/*v*) fetal bovine serum and 1% (*v*/*v*) antibiotics (100 U mL^−1^ penicillin and 100 μg mL^−1^ streptomycin).

### 3.6. Cell Viability Assay

Cell viability was assessed using the MTT colorimetric assay, which measures cellular dehydrogenase activity by reducing MTT to formazan. RWPE-1 and LNCaP cells were seeded at densities of 3 × 10^4^ and 1 × 10^4^ cells well^−1^, respectively, in 96-well plates and incubated for 24 h. Cells were then exposed to different concentrations (0.1, 1, and 10 μM) of compounds **1** and **2** for 24 h. MTT (2 mg mL^−1^) in distilled water was added to each well and incubated at 37 °C for 1 h. The formazan precipitate was dissolved in DMSO, and absorbance was measured at 550 nm using a microplate reader.

### 3.7. Western Blotting Analysis

RWPE-1 and LNCaP cells were seeded at 2 × 10^5^ and 6 × 10^5^ cells/well, respectively, in 6-well plates and incubated overnight. Cells were treated with TP (0.5 μM) for 1 h, followed by treatment with finasteride (10 μM) or compounds **1** or **2** at different concentrations (1 and 10 μM) for 24 h. Cells were washed with cold PBS, and cell lysates were prepared using lysis buffer containing a protease inhibitor cocktail. Protein content was quantified using the Bradford assay. Protein samples (30 μg) were separated by SDS-PAGE and transferred to a membrane. The membrane was blocked with skim milk and incubated with primary antibodies against 5αR2, AR, PSA, PCNA, and β-actin. After washing, immunoreactive bands were detected using peroxidase-conjugated secondary antibodies. Protein bands were visualized using ECL substrate and an imaging system.

## 4. Conclusions

The two new compounds **1** and **2** were isolated from a bioactive fraction of an association of two marine sponges of *Poecillastra* sp. and *Jaspis* sp. The compounds are commonly characteristic of the attachment of acyl chains on a zwitterionic moiety. The complete structure of **1** was established as (*Z*)-dec-3-enoyl (2*S*,3*S*)-epicapreomycidine based on the *J*BCA and ECD comparison. Although the absolute configurations of **2** were not assigned, the compound was revealed as a linkage of 2, 7*S**-dihydroxy-9*R**-methyloctadecanoyl group to the position of 2-amino of a tramiprosate moiety called homotaurine. The two compounds exhibited inhibitory effects of biomarkers (5αR2, AR, PSA, and PCNA) related to BPH on testosterone propionate (TP)-activated LNCap and RWPE-1 cells. In particular, compound **2** exhibited significant suppression of 5αR2 expression in two cell types, and it was superior to finasteride, a selective inhibitor of 5αR2. Additionally, considering the inhibition of the expression of other biomarkers closely related to BPH progression, these findings support the potential use of compound **2** as a therapeutic agent for the treatment of BPH.

## Figures and Tables

**Figure 1 marinedrugs-21-00491-f001:**
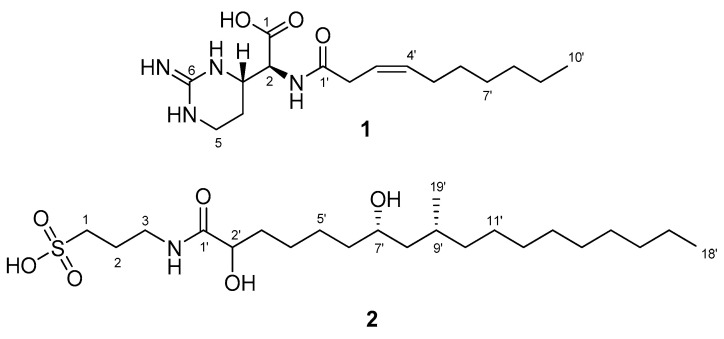
Structures of compounds **1** and **2** isolated from associated sponges.

**Figure 2 marinedrugs-21-00491-f002:**
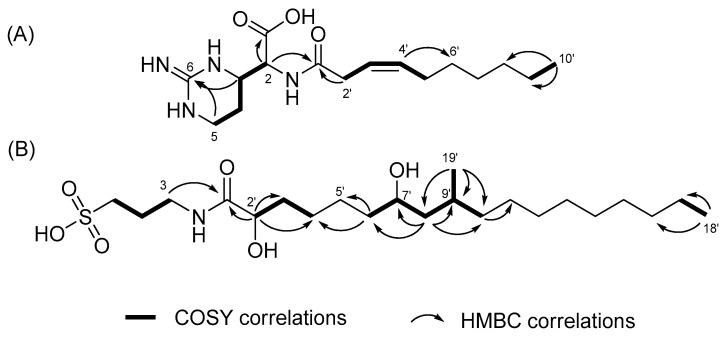
Key COSY and HMBC correlations of compounds **1** (**A**) and **2** (**B**).

**Figure 3 marinedrugs-21-00491-f003:**
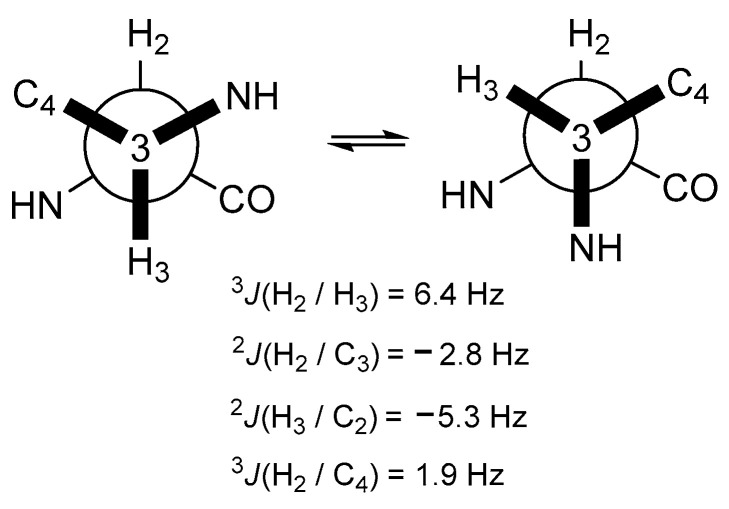
Relative configurations of C-2 and C-3 in **1** based on *J*BCA.

**Figure 4 marinedrugs-21-00491-f004:**
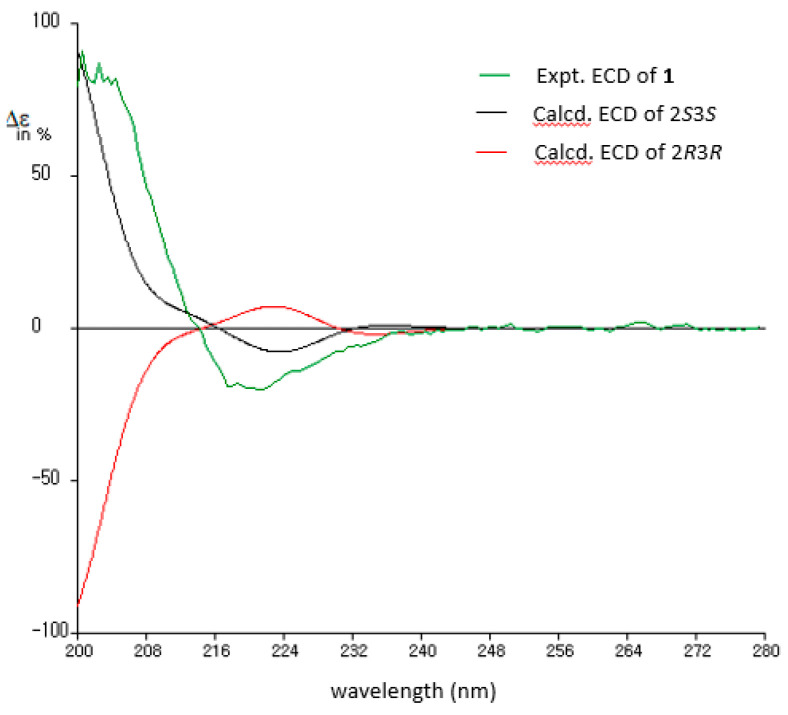
Comparison of experimental and calculated ECD spectra of **1**.

**Figure 5 marinedrugs-21-00491-f005:**
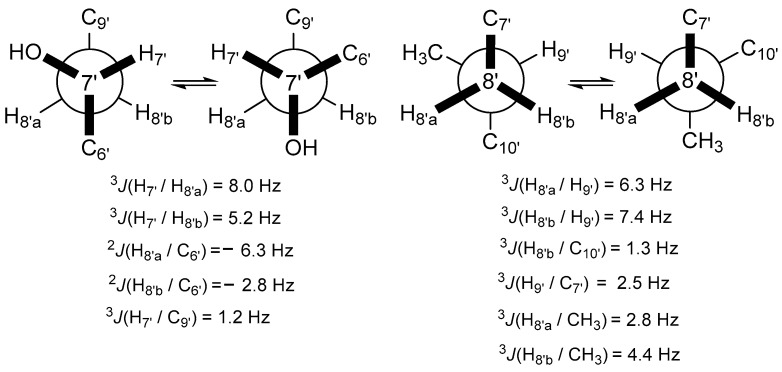
Relative configurations of C-7′ and C-9′ in **2** based on *J*BCA.

**Figure 6 marinedrugs-21-00491-f006:**
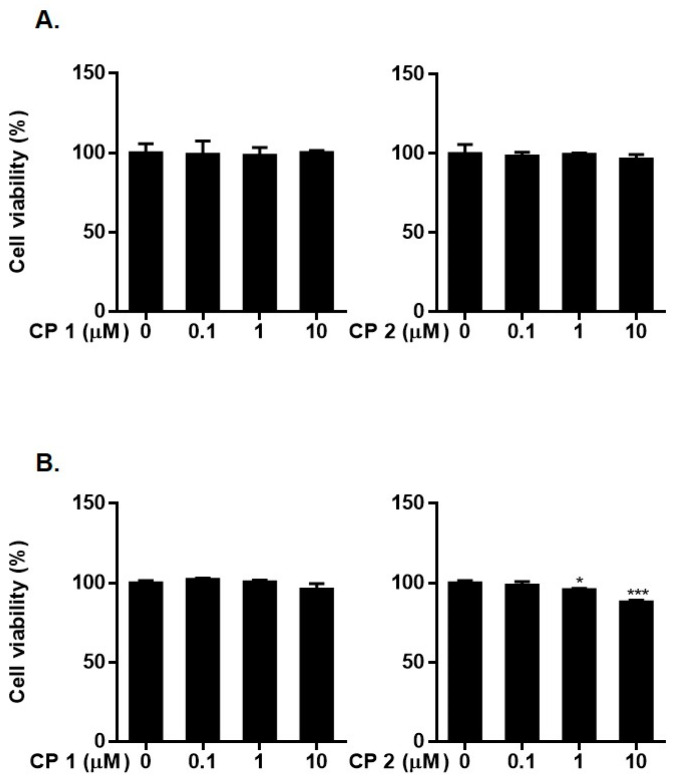
Cytotoxicity of **1** and **2** in LNCap (**A**) cells and RWPE-1 cells (**B**). Cells were treated with **1** or **2** (0.1, 1, 5, and 10 μM) for 24 h. Cell viability was determined by MTT assay. Results are presented as mean ± standard deviation (*n* = 3) * *p* < 0.05, *** *p* < 0.001 compared to non-treated controls.

**Figure 7 marinedrugs-21-00491-f007:**
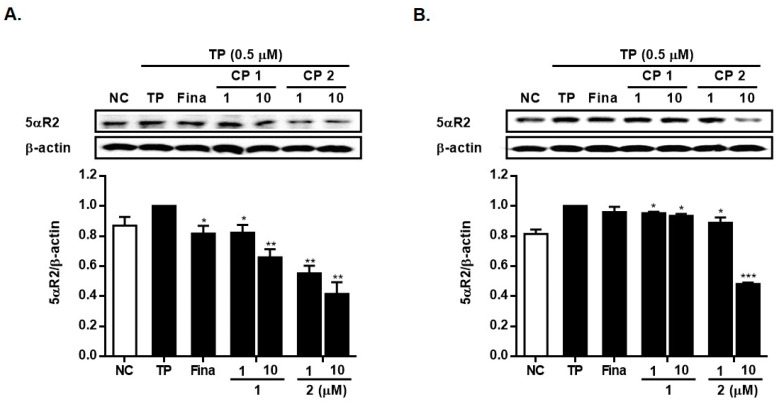
Inhibitory effects of **1** and **2** on the expressions of 5αR2 in LNCap cells (**A**) and in RWPE-1 cells (**B**). Cells were treated with testosterone propionate (TP, 0.5 μM) and **1** or **2** (1 and 10 μM) for 24 h. The expression levels of 5 αR2 were analyzed by Western blotting. Results are presented as mean ± standard deviation (*n* = 3); *p* < 0.01 compared to the non-treated control (NC), * *p* < 0.05, ** *p* < 0.01 and *** *p* < 0.001 compared to TP-treated cells.

**Figure 8 marinedrugs-21-00491-f008:**
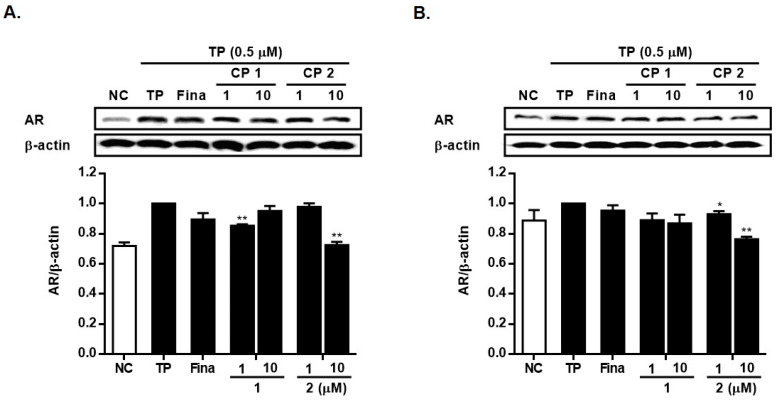
Inhibitory effects of **1** and **2** on the expressions of AR in LNCap cells (**A**) and in RWPE-1 cells (**B**). Cells were treated with testosterone propionate (TP, 0.5 μM) and **1** or **2** (1 and 10 μM) for 24 h. The expression levels of AR were analyzed by Western blotting. Results are presented as mean ± standard deviation (*n* = 3); *p* < 0.01 compared to the non-treated control (NC), * *p* < 0.05, ** *p* < 0.01 compared to TP-treated cells.

**Figure 9 marinedrugs-21-00491-f009:**
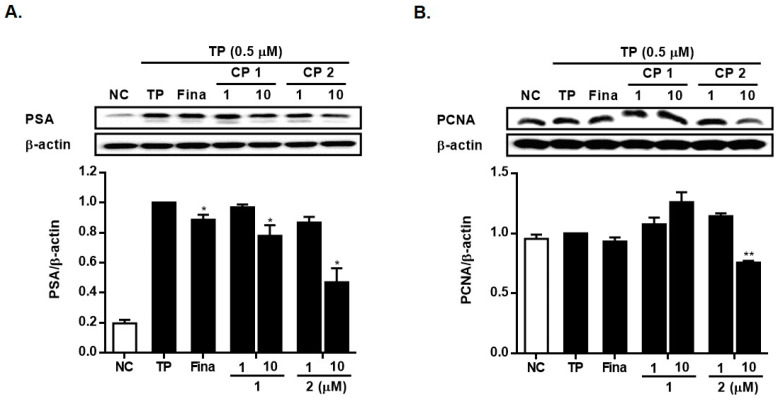
Inhibitory effects of **1** and **2** on the expressions of PSA in LNCap cells (**A**) and PCNA in RWPE-1 cells (**B**). Cells were treated with testosterone propionate (TP, 0.5 μM) and **1** or **2** (1 and 10 μM) for 24 h. The expression levels of PSA or PCNA were analyzed by Western blotting. Results are presented as mean ± standard deviation (*n* = 3); *p* < 0.01 compared to the non-treated control (NC), * *p* < 0.05, ** *p* < 0.01 compared to TP-treated cells.

**Table 1 marinedrugs-21-00491-t001:** The spectral data for **1** in CD_3_OD (^1^H for 500MHz, ^13^C for 125 MHz).

No	δ_C_, Mult	δ_H_, (*J* Hz)	COSY	HMBC
1 2 3 4 5 6 1′ 2′ 3′ 4′ 5′ 6′ 7′ 8′ 9′ 10′	174.5, C 57.6, CH 53.6, CH 23.2, CH_2_ 38.4, CH_2_ 155.5, C 174.7, C 35.6, CH_2_ 122.8, CH 135.1, CH 28.4, CH_2_ 30.5, CH_2_ 30.1, CH_2_ 32.9, CH_2_ 23.7, CH_2_ 14.4, CH_3_	4.39, d(6.4) 3.75, ddd(9.3, 6.4, 4.2) 1.75, m; 1.90, m 3.23, ddd(12.7, 10.3, 4.2) 3.40, dt(12.7, 4.65) 3.08, d(7.1) 5.55, m 5.61, m 2.10, dt(7.1, 6.9) 1.36, m 1.26~1.34 1.26~1.34 1.26~1.34 0.89, t(7.1)	H-3 H-2, H-4 H-3, H-5 H-4 H-3′ H-2′, H-4′ H-3′, H-5′ H-4′, H-6′ H-10′ H-9′	C-1, C-3, C-4, C-1′ C-1, C-2, C-4, C-5, C-6 C-2, C-3, C-4 C-3, C-4, C-6 C-1′, C-3′, C-4′ C-2′ C-2′, C-5′, C-6′ C-3′, C-4′, C-6′ C-4′, C-7′ C-8′, C-9′

**Table 2 marinedrugs-21-00491-t002:** The spectral data for **2** in CD_3_OD (^1^H for 500MHz, ^13^C for 125 MHz).

No	δ_C_, Mult	δ_H_, (J Hz)	COSY	HMBC
1 2 3 1′ 2′ 3′ 4′ 5′ 6′ 7′ 8′ 9′ 10′ 11′ 12′ 13′ 14′ 15′ 16′ 17′ 18′ 19′	38.8, CH_2_ 26.3, CH_2_ 50.0, CH_2_ 177.6, C 72.8, CH 35.6, CH_2_ 26.2, CH_2_ 30.6, CH_2_ 38.7, CH_2_ 70.3, CH 46.2, CH_2_ 30.6, CH 37.7, CH_2_ 28.0, CH_2_ 26.6, CH_2_ 31.1, CH_2_ * 30.8, CH_2_ * 30.5, CH_2_ * 33.1, CH_2_ 23.8, CH_2_ 14.5, CH_3_ 20.8, CH_3_	3.33, m 1.97, quint(7.8) 2.82, t(7.8) 3.98, dd(7.8, 3.9) ^a^ 1.57, m; ^b^ 1.74, m 1.43, m 1.33, m ^a^ 1.36, dt(7.1, 6.9); ^b^ 1.43, m 3.60, m ^a^ 1.27, m; ^b^ 1.34, m 1.60, m ^a^ 1.08, m; ^b^ 1.37, m ^a^ 1.27, m; ^b^ 1.35, m ^a^ 1.35, m; ^b^ 1.45, m 1.27~1.34, m 1.27~1.34, m 1.27~1.34, m 1.28, m 1.31, m 0.90, t(6.9) 0.89, d(6.9)	H-2 H-1, H-3 H-2 H-3′ H-2′, H-4′ H-3′ H-7′ H-8′, H-6′ H-7′, H-9′ H-8′, H-10′, H-19′ H-9′ H-18′ H-17′ H-9′	C-1′, C-2, C-3 C-1, C-3 C-1, C-2 C-1′, C-3′, C-4′ C-1′, C-2′ C-4′, C-5′, C-7′ C-6′, C-7′, C-9′, C-19′ C-10′ C-11′ C-16′, C-17′ C-8′, C-9′, C-10′

* Exchangeable carbons, ^a^ shielded proton, ^b^ deshielded proton.

## Data Availability

All data are contained within this article and Appendix A.

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
