# Peer review of "Two New Components from an Association of Marine Sponges Poecillastra sp. and Jaspis sp. and Their Inhibitory Effects on Biomarkers for Benign Prostatic Hyperplasia"

_marinedrugs, 2023, doi:10.3390/md21090491_

Round 1

Reviewer 1 Report

In this manuscript, Hwang and co-workers carried out a chemical investigation on an association of marine sponges Poecillastra sp. and Jaspis sp., which lead to the discovery of two new secondary metablites. Their structures were established by the extensive analysis of NMR, MS and IR spectra. Additionally, the absolute configuration of compound 1 was determined by ECD calculation. The two compounds exhibited inhibitory effects of biomarkers (5αR2, AR, PSA, and PCNA) related to BPH on testosterone propionate (TP)-activated LNCap and RWPE-1 cells. Based on these important and interesting findings, this work was suggested to publish in this journal.

However, based on the reviewer’s analysis of NMR and MS data, the structure of compound 1 was correct, but the establishment of the structure of compound 2 was not so solid. This made me gave Major Revision.

1. As shown in Figure S1, the detected mass data for [M−H] was 323.2087, not 323.2089.

2. P3L90&91: Please add the chemical shifts for ‘five protons within the mid-chemical shift range’, and revise the data ‘δH 1.30 to 1.50’ according to the records in Table 1. And revise the sentence ‘The other fragment encompassed protons at H 3.75 (H-3), 3.23 (H-5), 3.40 (H-5), 98 and 4.39 (H-2)(P3L98)’ accordingly.

3. The cross-peaks of H-3/H-4 and H-9/H-10′ were clearly observed in Figure S3, although H-9′ was overlapped. These peaks could be added in the third and last two rows of Table 1.

4. Please add the IR data for the arboxylic group in the corresponding sentence(P3L106).

5. Please finger out the HMBC correlation of H-2′/C-1′ in the corresponding sentence(P3L110).

6. The HMBC correlations of H-4′/C-2′ and H-4′/C-6′ were shown in Figure S6 in addition to H-4′/C-5′, but they were missing in Table 1. It was especially important to figure out H-4′/C-6′, which was the evidence for distinguishing two adjacent protons H-4′ and H-3′.

7. The HMBC correlations of H-3′/C-2′, H-4′/C-2′, H-4′/C-5′, and H-4′/C-6′ should be pointed out in the main text, which not only located the position of double bond, but also suggested a hexane chain linked to the double bond.

8. What was the configuration of double bond used for ECD calculation? In my opinion, to save time, it was better to assign the configuration before the ECD calculation.

9. It was better to show the chemical structures of three major fragments m/z 141 80, 138, and 166.

10. The COSY correlations of H-8/H-7′ and H-8/H-9′ were not shown for H-8′ in Table 2.

11. Please add some key COSY and HMBC correlations in the structure elucidation of compound 2.

12. Since the majority of 1H NMR signals of the long chain were overlapped, it was quite different to determine the numbers of methylenes between C-2′ and C-6′, and between C-6′ and C-8′ only based on the analysis of NMR data. Did you find some characteristic MS fragments to support the assigned structure of compound 2?

13. In my opinion, further support for the relative configurations of C-6′ and C-8′ was the NOE cross-peaks between H-6′ and H-8′ in the NOESY spectrum, not the cross-peaks between H-6′ and H-9′.

Others:

1. P1L28: ‘(2S; 3S)-capreomycidine’ → ‘(2S,3S)-capreomycidine’

2. Figure 1 caption: ‘compounds 1~2’ → ‘compounds 1 and 2

3. Figure 5 caption: ‘Relative configuration of C-2 and C-3’ → ‘Relative configurations of C-2 and C-3’

4. P10L269: ‘Poecillastra sp. and Jaspis sp.’ → ‘Poecillastra sp. and Jaspis sp.’

There were some grammar or typo errors. Some of them were given in the comments to the authors.

Author Response

We would like to thank for your careful review. Based on your points, we corrected the parts and enhanced the results. First of all, the structure of compound 2 was modified based on the extensive interpretation of the NMR spectra. Specifically, the HSQC-TOCSY spectrum was critical to determine the assignment of the carbons in the linear chain. We corrected the pointed part and added the data in the revised version. 

  1. As shown in Figure S1, the detected mass data for [M−H]was 323.2087, not 323.2089.

→ We corrected this mass data in the text.

  1. P3L90&91: Please add the chemical shifts for ‘five protons within the mid-chemical shift range’, and revise the data ‘δH1.30 to 1.50’ according to the records in Table 1. And revise the sentence ‘The other fragment encompassed protons at dH 3.75 (H-3), 3.23 (H-5), 3.40 (H-5), 98 and 4.39 (H-2)(P3L98)’ accordingly.

→ We added the five chemical shift values and revised the chemical shift range. We revised the sentence pointed.

  1. The cross-peaks of H-3/H-4 and H-9′/H-10′ were clearly observed in Figure S3, although H-9′ was overlapped. These peaks could be added in the third and last two rows of Table 1.

→ We added the COSY cross peaks which did not include in Table 1.

  1. Please add the IR data for the carboxylic group in the corresponding sentence(P3L106).

→ We added the IR absorption frequency (1626 and 3315 cm-1) in the text.

  1. Please finger out the HMBC correlation of H-2′/C-1′ in the corresponding sentence(P3L110).

→ We added the HMBC correlation in the text.

  1. The HMBC correlations of H-4′/C-2′ and H-4′/C-6′ were shown in Figure S6in addition to H-4′/C-5′, but they were missing in Table 1. It was especially important to figure out H-4′/C-6′, which was the evidence for distinguishing two adjacent protons H-4′ and H-3′.

→ In Table 1, we added the additional HMBC correlations related to the double bond.

  1. The HMBC correlations of H-3′/C-2′, H-4′/C-2′, H-4′/C-5′, and H-4′/C-6′ should be pointed out in the main text, which not only located the position of double bond, but also suggested a hexane chain linked to the double bond.

→ Along with addition of the HMBC correlations in Table 1, the HMBC correlations related to the double bond were described in the text.

  1. What was the configuration of double bond used for ECD calculation? In my opinion, to save time, it was better to assign the configuration before the ECD calculation.

→ The geometry of the double bond was explained after determining the position of the double bond. The information was placed in the last part in the original manuscript. And the Z-formed double bond was used in the calculation of ECD spectra.

  1. It was better to show the chemical structures of three major fragments m/z141 80, 138, and 166.

→ We presented the fragments on the chemical structure in Figure S9.

  1. The COSY correlations of H-8′/H-7′ and H-8′/H-9′ were not shown for H-8′ in Table 2.

→ We deleted the unobserved correlations after double checking.

  1. Please add some key COSY and HMBC correlations in the structure elucidation of compound 2.

→ We added the key correlations related to the double bond.

  1. Since the majority of 1H NMR signals of the long chain were overlapped, it was quite different to determine the numbers of methylenes between C-2′ and C-6′, and between C-6′ and C-8′ only based on the analysis of NMR data. Did you find some characteristic MS fragments to support the assigned structure of compound 2?

→ This point led us to recheck the structure of 2. Owing to the severely overlapped signals, the COSY, TOCSY, and HMBC correlations did not give a critical information, but the HSQC-TOCSY spectrum showed an additional carbon between C-6’ and C-8’. This was described in the text and the HSQC-TOCSY spectrum was added in the supplementary materials.

  1. In my opinion, further support for the relative configurations of C-6′ and C-8′ was the NOE cross-peaks between H-6′ and H-8′ in the NOESY spectrum, not the cross-peaks between H-6′ and H-9′.

 → In the relative configuration of C-6’ and C-8’ (newly, C-7’ and C-9’), the conformers are useful. From these conformers, the NOEs of H-6’/H-8’, H-6’/H-9’, and H-7’a/H-19’ were suggested and were observed in the NOESY spectrum. Among them, the NOE of H-7’a/H-19’ is very critical support for 8R* configuration.

Others:

  1. P1L28: ‘(2S; 3S)-capreomycidine’ → ‘(2S,3S)-capreomycidine’
  2. Figure 1caption: ‘compounds 1~2’ → ‘compounds 1 and 2
  3. Figure 5caption: ‘Relative configuration of C-2 and C-3’ → ‘Relative configurations of C-2 and C-3’
  4. P10L269: ‘Poecillastra sp. and Jaspis sp.’ → ‘Poecillastrasp. and Jaspis sp.’

→ We corrected the above points, but did not find the part for number 4.

Once again, we express our appreciation on your detailed points. We think that our result was enhanced thanks to your review.

Sincerely yours,

Reviewer 2 Report

The manuscript "Two new components of an association of marine sponges Poecillastra sp. and Jaspis sp. and their inhibitory effects on biomarkers for benign prostatic hyperplasia", shows relevant aspects of new compounds from sponges Poecillastra sp. and Jaspissp. However, authors should review the following aspects:

- The MTT assay determines the cytotoxicity of the compounds. However, it is not a specific test that indicates that the compounds have antiproliferative activity. The authors should include other experiments or indicate that using the MTT assay the compounds MAY be excellent antiproliferatives. And in the future this activity will be confirmed by other tests.

- In cytotoxicity assays, positive controls (drugs) must be added to compare the results obtained.

- In the discussions section we should further expand the section on biological assays. They must delve into the structure-activity relationship of the compounds on the pharmacological targets involved.

The manuscript presents several typographical errors resulting from writing in American and British English. Please review the entire manuscript.

Author Response

Thank you for your review. We have responses to your comments below.

  1. The MTT assay determines the cytotoxicity of the compounds. However, it is not a specific test that indicates that the compounds have antiproliferative activity. The authors should include other experiments or indicate that using the MTT assay the compounds MAY be excellent antiproliferatives. And in the future this activity will be confirmed by other tests.

→ In accordance with the reviewer's comments, we have revised the sentences in the results section describing the MTT assay results and briefly mentioned the possible activity of compound 2 on proliferation of prostate cells.

  1. In cytotoxicity assays, positive controls (drugs) must be added to compare the results obtained.

→ The reason for measuring the cytotoxicity of the compounds in this study through MTT analysis was to determine the non-toxic concentrations of the compounds before evaluating their efficacy in improving BPH. In previous studies assessing the cytotoxicity of marine toxins, okadaic acid was commonly used as a positive control; however, it was not applied in this study. Instead, finasteride was employed as positive control in the Western blot assay for evaluating the regulatory effects on enzymes related to BPH induced by compound 2. Please understand that obtaining additional toxicity results for the positive control in two prostate cell lines within the limited timeframe is practically challenging (considering the time required for cell stabilization and ensuring data significance through repeated experiments).

  1. In the discussions section we should further expand the section on biological assays. They must delve into the structure-activity relationship of the compounds on the pharmacological targets involved.

→ Thank you for the valuable comment. The reviewer suggested further elaboration on the structure-activity relationship among the compounds. In the present study, we employed activity-guided isolation techniques to isolate inhibitory compounds on 5-alpha reductase the association of Marine Sponges Poecillastra sp. and Jaspis sp. As shown in Figure 1, the isolated compounds 1 and 2 possess low similarity in structure, making it challenging to perform a structure-activity relationship (SAR) analysis at this stage. Based on the potent activity of these Marine sponges, our research team plans to continue to isolate the bioactive compounds and to perform SAR analysis when a sufficient number of compounds with structurally similarity are obtained.

Round 2

Reviewer 1 Report

The authors provided satisfactory responses to the questions raised by the reviewer in the first round of review. The manuscript can be accepted in the present form.

Just a reminder for Table 2: Because the data for CH2-14ꞌ were missing, the following data need to be adjusted.

Author Response

Thank your for your careful comment. In fact, the revised article has a corrected Table 2. Maybe you saw the tracking mode of the article. Table 2 was adjusted right as attached file. 

Reviewer 2 Report

The authors have made the corrections suggested by the reviewers.

Author Response

Thank you for your comments